# Methods to Account for Design for Disassembly: Status of the Building Sector

Carine Lausselet [1,*], Oddbjørn Andvik Dahlstrøm [2], Marit Thyholt [3], Aida Eghbali [4] and Patricia Schneider-Marin [4]

1 Department of Architecture, Materials and Structures, SINTEF Community, Høgskoleringen 7B, 7034 Trondheim, Norway
2 Asplan Viak AS, Postboks 24, 1300 Sandvika, Norway
3 Skanska Norge AS, Postboks 1175, Sentrum, 0187 Oslo, Norway
4 Department of Architecture and Technology, The Norwegian University of Science and Technology, Sentralbygg 1, 7491 Trondheim, Norway
* Correspondence: carine.lausselet@sintef.no; Tel.: +47-467-89-264

**Abstract:** Embracing the design for disassembly (DfD) mindset when constructing new and renovating existing buildings is a promising means of achieving our climate targets and putting the circular economy principles in practice, as promoted in the European Green Deal. Current greenhouse gas emissions' accounting frameworks only deal with DfD to a certain extent. A better and more common understanding of how this can be carried out will not only help promote DfD but also shed light on how DfD should be seen in the context of other emission reduction measures. This could help to achieve balanced and credible scenarios that can be used in policy-making processes. When building components or materials are used over several cycles (buildings), the allocation of environmental impacts across the different buildings must be discussed. In an attempt to address this issue, this study examined whether and how current LCA standards for construction products and buildings consider such allocation issues.

**Keywords:** design for disassembly (DfD); circular economy; life-cycle assessment (LCA); building





## 1. Introduction

Following the launch of the European Green Deal [1], which makes sustainable products the norm in the European Union, increases circular business models, and empowers consumers for the green transition, the circular economy (CE) has gained considerable attention in policy-making processes. One of the proposals in the Green Deal aims to strengthen the internal circular market for construction products and ensure that the regulatory framework in place is suitable for ensuring that the built environment meet the sustainability and climate goals.

The Green Deal and CE strategies support holistic approaches. For buildings, this means that the focus shifts from a sole focus on energy efficiency measures [2] to widen the scope and shed light on material-related greenhouse gas (GHG) emissions from construction and renovation activities [3]. Such comprehensive approaches are key to fulfilling the obligations under the Paris Agreement and to ensuring that the global average temperature does not rise to 1.5 or 2 °C above preindustrial levels [4].

An economy that closely aligns with circular targets has the potential to contribute to value creation and jobs based on new business models that offer repair, rental, and sharing [5]. The focus on CE strategies is strong at both European and national levels. Norway is no exception [6] and actively works on implementing the CE Action Plan with a focus on the following six strategies, given in order of importance: 1. Reconsider, 2. Reduce, 3. Reuse, 4. Repair, renovate, and reproduce, 5. recycle and utilize residual raw materials,

and 6. energy recovery. Those CE strategies are designed to increase the durability, repair, leasing, and rental of consumer goods.

The EU CE Action Plan emphasizes the importance of "getting the economics right" and the importance of acting at the community level [7]. For the case of the building sector, this framework translates into planning and building for future disassembly and the re-use of material [8]. In the international literature, "Design for disassembly" or "Design for Deconstruction" (DfD) are the terms used for these processes [9]. Yet, quantitative data on building materials' reuse, recycling, and deconstruction activities are scarce [10].

DfD has also been highlighted as an important measure for a more circular building sector, as part of the new EU taxonomy [11]. A definition of DfD for construction works is given in the ISO standard 20887:2020 "Sustainability in buildings and civil engineering works—Design for disassembly and adaptability—Principles, requirements and guidance" [12] as "an approach to the design of a product or constructed asset that facilitates disassembly at the end of its useful life, in such a way that enables components and parts to be reused, recycled, recovered for energy or, in some other way, diverted from future waste stream". At the national level, in Norway, it also became clear in June 2022 that the revision of the technical building regulations TEK 17 [13] will require buildings to be designed for later dismantling, in line with the DfD principles.

A life-cycle assessment (LCA) is a useful method to assess resource use and emissions over the entire life cycle of a product or service [14]. In the context of the built environment, an LCA allows for the quantification of the environmental sustainability of buildings [15]. Interesting methodological questions arise when combining CE principles with an LCA. For example, if an LCA allows one to show the advantages of the CE in environmental terms, companies or societies could define targets based on the LCA results [16].

A potential benefit of DfD, in addition to resource savings, is the decreased life-cycle environmental burdens induced by avoiding raw material extraction and product manufacturing. However, how those potential environmental benefits can be accounted for has not been clearly defined in the laws, regulations, and/or standards, and hence must be investigated and clarified. The question "(How) do current laws and regulations facilitate that DfD can be considered in GHG accounting in the building sector?" shall be addressed. By addressing this question, the building sector will be in a better position to lay down guidelines for how the rules can be applied to promote DfD in new building and rehabilitation projects, potentially reduce future climate impact and resource use, and minimize design and construction costs.

The aim of this study is to assess existing methods that address the accounting and allocation of GHG emissions of DfD design concepts, and to analyze whether these can be used or further developed. The investigation serves as a basis for testing the influence of calculation methods on future representative building case studies. This study is part of the research project "SirkBygg—Circular new buildings—Design and construction for dismantling and reuse" [17]. SirkBygg aims to make it easier and more affordable to build for future dismantling and reuse and thereby contributes to the Sustainable Development Goals (SDGs) 9. "Industry, innovation and infrastructure", 11. "Sustainable cities and communities", and 13. "Climate action".

This study is organized in the following manner. The CE is first set in the context of the building sector in Section 2. Section 2 is then further developed in terms of the limitations of the implementation of circular economy principles in the building sector (Section 2.1) and set in the context of the Norwegian building sector (Section 2.2). How to account for circularity in LCA frameworks is then addressed in Section 3, in current standards (Section 3.1), in terms of allocation of materials use and reuse (Section 3.2), and as tested in case studies (Section 3.3). The LCA limitations for accounting for the benefits of circularity in terms of DfD are then presented in Section 4, including the valuation of future avoided emissions (Section 4.1) and the choice of a time horizon (Section 4.2). Biogenic carbon is then discussed in Section 5 and carbon in concrete in Section 6, in light of the DfD principles. A closer look is taken at the existing DfD accounting practices in the Norwegian

building sector in Section 7, before finishing with Section 8, which presents the conclusions and future work.

## 2. Circular Economy in the Building Sector

The building sector had the highest share by weight of waste, with 37% of waste generation, compared to all the other economic activities in Europe in 2022 [18]. The building sector contributes to the building stock that is in a continuous state of growth, placing pressure on resource consumption, related contributions to GHG emissions, and planetary degradation [19]. CE principles would thus help alleviate this risk. However, the application of CE principles to the building sector is limited [20]. Shedding more light on the possibilities for a CE in the building sector to transition to a more circularly built environment is thus crucial [21].

To ensure the optimal implementation of the CE principles in the building sector, it is important to embrace a holistic manner and combine the correct selection of construction materials with the best building design and choice of building products. One example is the Nordic guide to sustainable materials [22] that introduces circular criteria for choosing materials as follows: (1). long service life of materials, (2). a low maintenance need of materials, (3). easy repair of materials, (4). recyclability of materials, (5). Reuse of materials, and (6). low environmental impact during service life.

### 2.1. Limitations of the Implementation of Circular Economy Principles in the Building Sector

The CE is still regarded as a complex and new paradigm that requires a clearer roadmap to be implemented in the building sector [21]. One reason behind this complexity is that the CE frameworks are site-specific, since they depend on a variety of environmental and economic factors, including building components and materials, transportation, and the political and economic contexts.

Another constraint to developing a CE in the building sector is the lack of related research in this field. Most of the research in Europe has been conducted on waste management efficiency rather than waste reduction or reuse, which has boosted the rate of downcycling [23]. This research gap has also resulted in limited data streams and indicators across the globe and the Nordics, particularly for the CE's inner loops, which include strategies such as reducing, extending product life-cycles, reusing, and refurbishing [24].

When DfD is applied, building elements are designed in a manner that allows for the different parts to easily be taken apart at the end of their useful life so that they can be diverted from the waste stream and reused, either directly or through material recovery. In addition, the use of DfD as a CE strategy will increase the adaptability, durability, and reusability of products while lowering the risk of damage and loss of value for subsequent life cycles [25]. However, despite DfD being recognized and promoted as a low-carbon CE service-life-extension technique, the main barriers are not technological but lie in the adoption of the DfD principles by the building sector along the whole supply chains and markets [26].

### 2.2. The Norwegian Building Sector

The Norwegian building sector is the largest single source of waste, with waste from the construction, rehabilitation, and demolition of buildings that accounted for 25% of a total of 12 million tons of waste in 2021, as shown in Figure 1 [27]. From this total, 55% are sent to material recovery (i.e., bricks and concrete and other heavy building materials, 48%; asphalt, 18%; metals and wood, 10% each), 19% to incineration with energy recovery (i.e., mixed waste, 57%; wood waste, 39%) and 23% to landfill (i.e., bricks and concrete and other heavy building materials, 44%; polluted bricks and concrete, 38%). On the other hand, the European Waste Directive stipulates that at least 70% of nonhazardous construction and demolition waste needs to be recovered starting in 2020 [28].

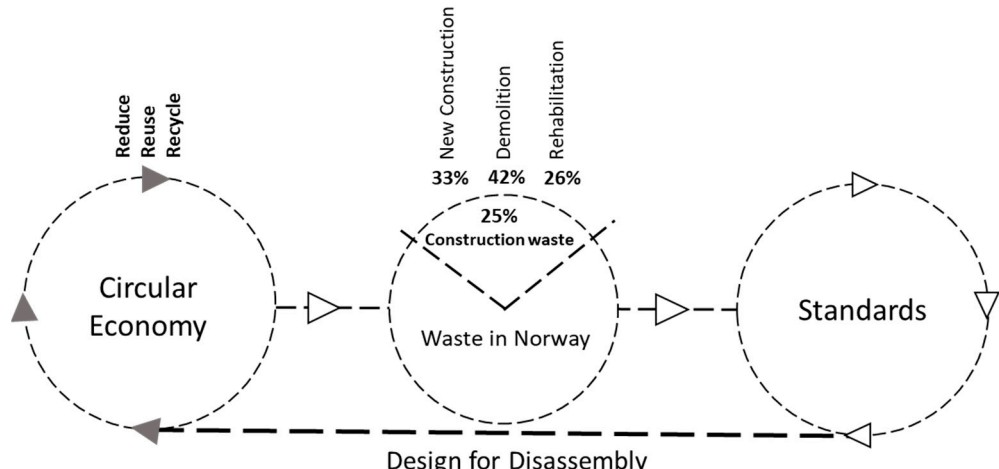

**Figure 1.** Implementing circular economy strategies in the building sector by means of standards.

The Norwegian figures fall short of this goal of 70%. An increase in the material recovery rate in waste streams could be accomplished through advanced waste sorting, which requires the careful disassembly of building components and enables the reuse of waste as a resource. The Norwegian building sector thus holds a unique opportunity to increase its circularity rate by using reused and reusable materials, e.g., using DfD. In addition, selecting products that are suitable for reuse and recycling will help to fulfil the new requirement of the building construction standard TEK 17 [13], which imposes material use requirement.

According to the Platform for Accelerating the CE [29], several core sectors and standards have been developed with the intention of bridging the circularity gap in the building sector. Three of them have the potential to be key change agents in the Norwegian CE landscape: design for the future, sustain and preserve existing buildings, and utilizing waste as a resource. The goal of these suggested strategies is to slow material flows by extending the service life of building components and to close loops through reversible construction design and smart material management.

## 3. Accounting for Circularity in LCA Frameworks

A number of circularity indicators are under development for construction works. The Urban Mining Index assesses the circularity potential of a building at different levels: building, building component, construction element, component layer, material, and raw material [30]. The material circularity indicator, developed by The Ellen MacArthur Foundation and Granta Design, allows for companies to identify additional, circular value from their products and materials, and thereby mitigate risks from material price volatility and material supply [31]. However, the LCA is the most used method to evaluate the CE potential in the building sector [32].

### 3.1. Circularity in Current LCA Standards

To ensure the implementation of CE strategies in the building sector, LCA standards should be in place to ensure the harmonization of GHG accounting.

LCA standards for assessing the environmental performance of construction works consist of:

- EN 15978:2011 [33], which provides standard instructions for assessing the environmental performance of the CEN TC 350 sustainability of the construction works' standard family.
- EN 15804:2012 [34], which provides instructions for the Environmental Product Declaration content in the CEN TC 350 sustainability of the construction works' standard family.

- NS 3720:2018 [35], specifying calculation rules for GHG accounting for buildings in Norway.

Per these standards, the LCA for buildings divides the life cycle into four stages: the production stage (A1–A3), the implementation stage (A4–A5), the use stage (B1–B7/B8), and the end-of-life stage (C1–C4), each consisting of distinct modules. In addition, a last module (D) expresses the net benefits and burdens outside of the system boundary. Module D considers further stages beyond the building's system boundaries by including the reuse and material recycling of building products that are used in new buildings or other applications.

Using a CE approach, the linear model of the LCA can be redesigned as a loop, as shown in Figure 2.

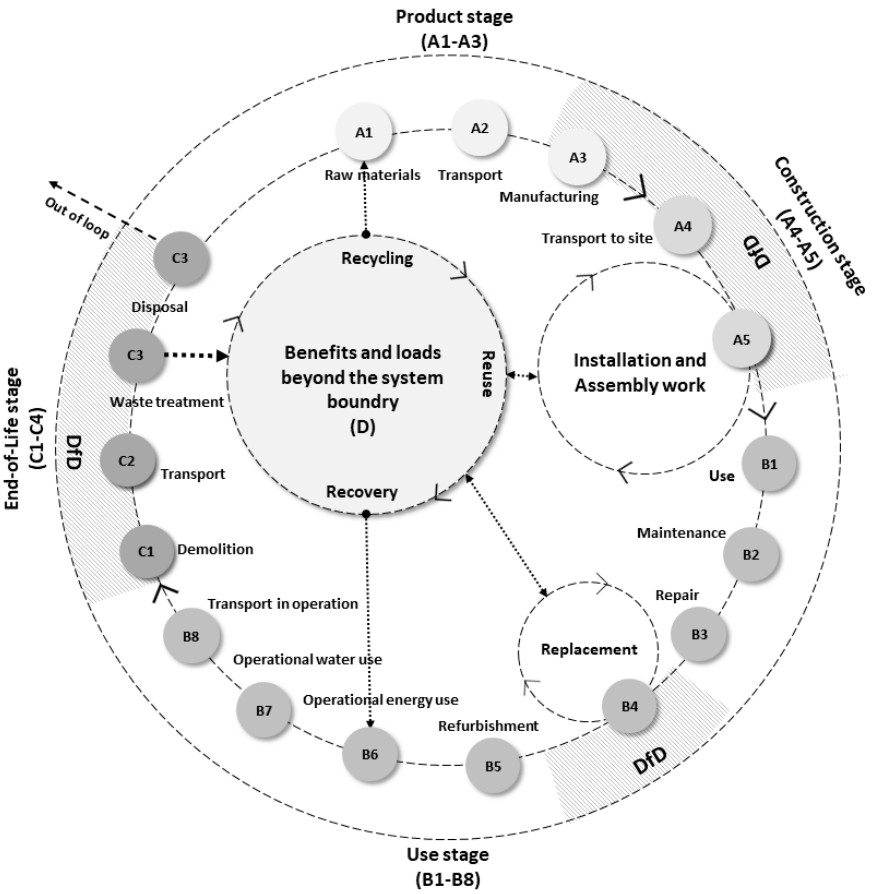

**Figure 2.** Life-cycle stages: circular approach and the implication of DfD in the LCA.

Module D can further be allocated to consequences beyond the system boundary that apply to energy and/or materials. "Module D—Energy" has been "regulated" and quantified using the Norwegian standard NS 3720:2018 [35]. "Module D—Materials" recognizes DfD by considering possible material recycling and reuse but does not quantify these processes.

In Norway, module D has mostly been used in connection with energy, as a consequence of the zero-emission building and zero emission neighborhoods [36] definitions, based on their net-zero GHG emissions balance regarding the excess energy produced on-site that can be sent to the grid. The surplus energy produced on-site was assigned GHG emissions credits. This GHG emissions accounting method was standardized in the Norwegian standard NS3710:2018 "Method for greenhouse gas calculations for buildings" [35].

The surplus energy is calculated on an hourly or annual basis with two different scenarios for future GHG emissions linked to electricity production, scenario 1 (Norway) and scenario 2 (EU28 + Norway), based on a Norwegian and European production mix,

respectively. In practice, scenario 1 regards Norway as an isolated electricity system without the import/export of electricity, whereas scenario 2 assumes that electricity flows freely between European countries, including Norway.

In the same way as two future scenarios for electricity are defined in NS3720:2018 [35] and reported in module D, scenarios for the future avoidance of GHG emissions when "exporting" reused or recycled materials in another building should be defined. Defining such scenarios is currently left to the individual projects and/or industry schemes.

### 3.2. Allocation of Materials Use and Reuse

The methods for allocating materials to a particular building (A1–A4), out of the building (C1–C4 and module D) applicable to several cycles (buildings), can roughly be grouped into three main approaches [37]: 100:0 ("cutoff"), 0:100 ("end-of-life recycling"), and 50:50 ("equal share"). These allocation methods favor either incoming (100:0) or outgoing (0:100) secondary materials. To provide a broader range, the European Commission has developed the "Circular Footprint Formula" as part of the "Product Environmental Footprint (PEF)" [38], which accommodates multiple allocation options by covering recycled content on the input side and material recovery and reuse at the end of the first cycle, by using an A-factor that reflects the market situation (i.e., whether there is a high or low supply and demand for the material) and the change in quality between cycles. The mentioned allocation methods are described in the subsections below.

**100:0** is also called "cutoff" in the recognized LCA database Ecoinvent [39]. "Cutoff" means that the "division" between the first and the second user is at the "end-of-waste", as defined in the annex to EN 15804:2012 [34]. The method is described in EN 15804:2012 [34] and EN 15978:2011 [33]. The environmental contribution of the production stage (A1–A3) is allocated to the first user (first building). The next user (next building) bears the environmental contribution in A1–A3 related to the use of recycled materials or the reuse of resources in the first building in which the environmental contribution takes place in module C3. The method encourages actors to use recycled and reused materials (gain in A1–A3) but does not provide any environmental (to be understood as GHG emissions) benefit by designing for future dismantling and reuse for use in the next building. *The method therefore provides no incentives for DfD. This 100:0 approach is standardized in EN 15804:2012 [34], EN 15978 [33], and NS3720:2018 [35].*

**0:100:** The environmental contribution of the production stage (A1–A3) is allocated to the last user (building). The method encourages actors to design for future dismantling and reuse for use in the next building but does not provide an incentive to use recycled materials or reuse already-existing building components. *The method therefore provides incentives for DfD. However, considering the relatively long lifetimes of building components, scenarios for reuse and material recycling are difficult to accurately predict. This 0:100 approach is not standardized.*

**50:50:** The environmental contributions from the production stage (A1–A3) are allocated between the first and the last user. *The method therefore provides some incentives for DfD. This 50:50 approach is not standardized.*

**A factor:** This approach, recommended by the European Commission [38], reflects the market realities and distributes the environmental contributions and gains from material recycling and new material production between the supplier and user of the recycled material and/or reused components. A = 1 reflects a 100:0 approach (i.e., credits are given only to the recycled content), whilst A = 0 reflects a 0:100 approach (i.e., credits are given only to end-stage recyclable and reusable materials). A is preferably in the range $0.2 \leq A \leq 0.8$ to always capture both the aspects of recycling and reuse at the end of the building's lifetime. The new version of EN 15804 + A2:2019 [40] includes a better harmonization between the PEF and environmental product declarations (EPDs). The main difference between the PEF and the EN 15804 + A2:2019 [40] approach used in EPDs is the level of aggregation. While the PEF simplifies the information for consumers by weighting, aggregating, and introducing a type of labeling scheme, the EPD provides non aggregated information for professional users. The PEF and EPD are now harmonized in terms of

indicators, international reference life-cycle data system (i.e., ILCD data format [41]), and details describing their use, emissions, and content of biogenic carbon.

*3.3. Testing of the Allocation Methods in Case Studies*

The authors of [42] tested the effect of three of the allocation approaches named above (100:0, 50:50, and A factor) on the LCA results and the subsequent incentive they provide to the building sector. Two "circular building components" with different characteristics were used: (1) a long-life concrete column designed for direct reuse with a high(er) uncertainty in subsequent reuse and (2) a short-lived recyclable roof felt with less uncertainty about future recycling, given the shorter time horizon. The LCA results across the allocation methods all allocated the highest impact to the first user. However, because the "A factor" allocation method considers the change in material quality over each use cycle, the results from the "A factor" allocation method allocated a greater impact to the first user of the shorter-lived building product than to the first user of the longer-lived building product.

The authors of [43] investigated how current calculation practices from European standards for an LCA on buildings affected building design when the CE and material loops were the focus, with two case studies: one building built from reused/recycled materials and a second building built with DfD principles. The results showed that DfD was not credited with any environmental gains (i.e., GHG emissions) when using the European framework (EN 15978:2011 [33] with corresponding NS3720:2018 [35]) as it stood then. Interestingly, the results showed that the building elements with a short lifespan received the greatest environmental benefit from the use of DfD.

Another case study by [44] used several GHG emissions' allocation methods to assess the climate contribution of 118 buildings in Winterthur, Switzerland, with three cycles: first (building), intermediate (building), and final (building). Their results varied across the allocation methods and assumptions made in connection with the building components allocated to the first, intermediate, and final cycles (buildings). Interestingly, all the allocation methods used led to similar results for the intermediate building, which were found to be smaller than those of the first and last building, meaning that, when averaged over all allocation methods, reuse always lead to favorable figures if building components were said to belong to the intermediate building. Some life-cycle stages (the implementation stage A4−A5, and the use stage B1−B5) gave the same results regardless of the allocation method used, whereas other life-cycle stages (the product stage A1−A3, the final stage C1−C4, and consequences beyond the system boundary D) showed large variations across the allocation methods. The study pointed to the need to include several critical functions that are currently hardly quantifiable, such as built-in utility value, versatility, storage and transformation impacts, user-owner separation, demountability, and design complexity.

## 4. LCA Limitations for Accounting for the Benefits of Circularity in Terms of DfD

The LCA framework and indicators also seem limited when considering the broader context and network of CE resource flows. According to [32], LCA indicators shall be complemented by CE indicators to reflect the complexity of CE resource flows because the LCA was not developed to optimize circularity, which can have environmental qualities beyond the indicators analyzed by the LCA. Additionally, to conduct a comprehensive LCA when exploring the environmental benefits or trade-offs of reuse and recycle practices, as it is the case when using DfD as a CE strategy, several environmental impact categories shall be considered [45]. This is to avoid problem-shifting, i.e., solving one environmental issue by creating another.

To include the CE in the LCA and consider the benefits of DfD in the final stages of the LCA (C–D) would be problematic for two reasons. Firstly, it is difficult to standardize the crediting of reuse (D) and material recycling (C3). Second, the approach is adapted for one building or one use-cycle. For several buildings (several cycles), as in a CE context, certain cycles can be omitted because they are not covered by the assessment. These two aspects are explained further below.

In EN 15978:2011 [33] and NS 3720:2018 [35], the building is considered a composition of various building products (components, materials, and structures). These may have different service lives for technical or functional reasons. However, the standards do not lay down guidelines for how different building products not built independently of each other should be considered, and replacement (preferably for reuse) may mean that other building products, with an initially longer lifespan, must also be replaced [46]. This issue is relevant to the exchange of materials during renovation (module B4 or B5). The multilayered disassembly sequence planning method [47], which attributes a property in terms of, e.g., disassembly attributes such as liaison and geometric feasibility to the product bill of material could also help in bridging this issue.

According to EN 15978:2011 [33] and NS 3720:2018 [35], consequences related to future reuse, material, and energy recovery outside the system boundary must be calculated in module D and reported separately. However, it is also not clear how module D will affect the overall GHG emissions calculations. In most GHG emissions calculations for buildings, some of the modules between A1 and C4 are omitted, either because they were deemed not relevant or because of data gaps [48]. Module D is a challenge in the calculations as it should show the effect of replacing a material or product in the near or far future. If, e.g., a building element includes the emission factor for module D in an EPD, as described in EN 15804:2012 [34], this can indicate the reduced GHG emissions obtained by replacing a new building element. However, there is a great level of uncertainty when determining GHG emissions derived from producing any new building element (and materials) 50 or 60 years from now. Additionally, it is questionable whether emissions that will be avoided in the future, through module D, should be emphasized as much as the focus on reducing emissions from the production of materials and the construction phase today (A1–A5). There is also a risk of double-counting the reduction in GHG emissions if a building includes reduced or avoided emissions in module D in the GHG emissions account, and the next building includes reduced emissions in A1–A3 by using the same materials.

*If module D is to be included in the emissions calculations, it should be clearly defined what the products will replace in the future when they are dismantled and reused.* Current standards or regulations are not clear on this aspect, meaning that this assessment is left to the individual project or EPDs. EPDs specify module D for some products based on current practices regarding waste treatment, as set out in product category rules (PCR) for the product under consideration. In practice, this means that EPDs usually consider module D because of energy recovery, not material recovery or reuse.

### 4.1. Valuation of Future Avoided Emissions

DfD will be able to contribute to the avoided GHG emissions and reduce new resource consumption in the future. DfD will enable the replacement of new materials, but the climate effect of this process depends on the carbon intensity of the replaced materials at the time the building is to be dismantled and the building products reused. *In the years ahead, it is natural that an increased focus on material recycling and an increased degree of recycled content in new materials, as well as more climate-efficient production processes, will contribute to a lower carbon footprint for new building products. The substitution effect of DfD in GHG accounting should therefore take this into account.*

There is a significant focus on achieving the targets of the Paris Agreement at present, and reducing GHG emissions today can, if the Paris Agreement's targets are the premise, be given greater weight than emissions in the future. This consideration is relevant to include in a GHG account.

In NS 3720:2018 [35], future emissions—or avoided emissions in the case of, for example, solar power production and export from buildings—are calculated with a technology factor that is based on the development of emission intensity in the European power grid (for scenario 2 (UE28 + NO). *A corresponding technology factor for materials is lacking in the standards at present.*

*4.2. Choice of Time Horizon*

Some GHGs, such as $CO_2$, stay in the atmosphere for a long time. The standard EN 15804:2012 [34] specifies that a time horizon of 100 years (GWP100) must be used when preparing EPDs. Since NS 3720:2018 [35] refers to EN 15804:2012 [34], GWP100 must also be used in GHG calculations for buildings. When using a time horizon that is shorter than 100 years, for example by setting climate targets in 2030 and 2050, the $CO_2$ that is emitted today should have more time to heat up the atmosphere than $CO_2$ that is emitted, for example, in 2037. Therefore, for a given target at a certain point in time, it is correct to say that the $CO_2$ emitted today is more potent and will contribute to a more accumulative climate effect than emissions that occur far in the future (up to 100 years) when a 100-year time horizon (GWP100) is considered. This means that emissions that occur (or are avoided) can (and should?) be weighted more today than (avoided) emissions sometime in the future. *This type of weighting with respect to DfD is not currently found in any standard or other calculation regulations for GHG calculations for buildings.*

## 5. Biogenic Carbon

Converting building components into carbon sinks to store carbon during their service life is one way to implement resource efficiency in the building sector. The term "Bioeconomy Strategy" was first used in 2012 to refer to the production of renewable biological resources and their potential for conversion into construction components [49]. A possible life cycle of renewable materials and cascade use of biomaterials is shown in Figure 3. However, a clear consensus on how to model the biogenic carbon released or absorbed during the lifetime of a building is lacking [50]. After a thorough systematic literature review, Andersen et al. [51] found that the majority of wooden-building LCAs applied similar methods and often left out biogenic carbon from the assessment or simply did not declare it.

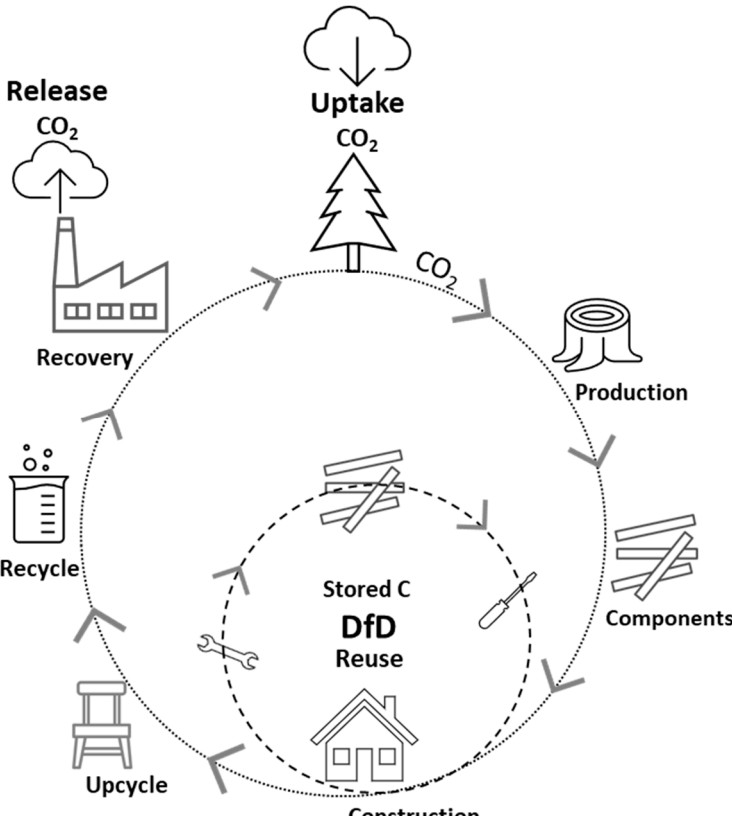

**Figure 3.** Life cycle of renewable materials and cascade use of biomaterials.

### 5.1. Wood as a Trend Material in the European Market

In Europe, wood-based construction materials constitute a substantial stock of construction materials, with the main driver being the alignment with the European climate targets [52]. The other benefits of wood-based materials are their high potential for recycling, reuse, and energy recovery at their end of life [53]. From a CE point of view, even agricultural and industrial wood wastes can play an important role in CE, as they can serve as a valuable secondary resource for making other products, e.g., insulation or chipboard panels. The reuse potential is considered an advantage in the circular scheme as biogenic carbon could be stored longer as it would be kept in a closed cycle. That is, wooden products such as insulation panels hold a tremendous potential to act as a carbon sink when used in new constructions, renovations, or the retrofitting of old buildings [54].

The issue with the use of wooden materials in a CE framework is that if wood is contaminated with harmful chemicals, reusing or even recycling that wooden component is not possible [55]. These chemicals are mostly used on wooden components to improve their performance regarding fire and humidity. The other challenge with reusing wooden products is that the quality, durability, and resistance to fire depends on the average characteristics of reclaimed materials.

### 5.2. Uptake of Biogenic Carbon in Standards

Biogenic carbon is carbon stored in biomass (tar) and soil via photosynthesis. In EPDs, according to NS 15804:2012 [34], the absorption and emission of biogenic carbon is calculated according to NS-EN 16485:2014 "Round and sawn timber—Environmental Product Declarations—Product category rules for wood and wood-based products for use in construction" [56]. This is based on the modularity principle in EN 15804:2012 [34], where emissions must be accounted for in the module in which they occur.

The amount of carbon dioxide is calculated in accordance with NS-EN 16449:2014 "Wood and wood-based products—Calculation of the biogenic carbon content of wood and conversion to carbon dioxide" [57]. As a standard, the content of biogenic carbon is calculated from the weight of dry wood. It is further assumed that 50% of the biomass in dry wood is carbon. Furthermore, the ratio between the atomic weight of carbon (12 g/mol) and $CO_2$ (44 g/mol) is used to convert the uptake of biogenic carbon into an uptake of $CO_2$.

Biogenic carbon must be reported in the module in which the absorption/discharge takes place, which means that the absorption of biogenic carbon is in A1, while emissions from waste management (burning) are in C3. Because the current waste management for construction wood in Norway is waste-to-energy [27], stored biogenic carbon will be released during waste management. A common practice in the GHG emissions' calculations has been to assume that the uptake of biogenic carbon in the growth phase (A1) will be equivalent to the emissions of biogenic carbon during combustion (C3), when *immediate oxidation* (carbon-neutral cycle) of the carbon is assumed [50]. This assumes sustainable forestry where new growth in the forest is equal to or greater than felling.

If it is possible to avoid sending wood to waste-to-energy when the wood is to be disposed of in the future, the emission of biogenic carbon during incineration can be avoided. Depending on what is done with the wood after use, the wood can therefore function as a future *store* of biogenic carbon if the wood product is not burned (in facilities without carbon capture) or composted. This factor comes into play in GHG emission calculations for buildings, building elements, and products when they are designed for disassembly and reuse. The stored biogenic carbon in the product will still be stored in the *next life cycle* of use. The challenge that is not stipulated in the standardization is how this should be included in the GHG accounts. There will be a similar effect when, in the future, the wood product is treated as waste in incineration plants with carbon capture and storage, which means that biogenic carbon in the wood that is normally released during combustion can be captured and not released into the atmosphere.

*5.3. Limitations*

It is currently not common practice to include the effect of carbon storage in wooden building elements in GHG emissions calculations for buildings in calculations that follow NS3720:2018 [35]. *That is, no clear guidelines have been found in standards and methods that answer the challenge of calculating the GHG emission reduction effect of DfD when the calculations include the uptake of biogenic carbon.* The deficiency in the current calculation rules is how stored carbon (negative emissions) should be handled during the transition from one life cycle to the next (from one building to the next building), when, in the current standard, it is always assumed that wood products are incinerated after the end of the life cycle. Today's calculation rules assume, as a simplification, that there is a neutral cycle for biogenic carbon, where absorption is the same as emission regardless of time, which does not agree with reality.

## 6. Carbon in Concrete

*6.1. Carbonation of $CO_2$ in Concrete*

In the production of cement, a mixture consisting mainly of limestone ($CaCO_3$) is burned with other raw materials such as quartz, clay, and slate. First, the mixture is crushed, and then heated in large, rotating furnaces to a material temperature of around 1450 °C. This starts a chemical process (calcination) where carbon dioxide ($CO_2$) is driven away from the limestone, and reactive calcium oxide (CaO) is formed, as shown in Equation (1).

$$CaCO_3 + heat => CaO + CO_2 \tag{1}$$

$CO_2$ must be removed from the limestone for the cement to acquire the properties that make it possible to make concrete that hardens.

Hardened concrete reacts with air, and this starts an aging process called carbonation. In the process, $CO_2$ from the air dissolves with the pore water in the concrete. $CO_2$ from the air is chemically bound by the formation of stable $CaCO_3$. This also means that the pH in the pore water drops below 10. This is the opposite process of calcination, as shown in Equation (2).

$$Ca(OH)_2 + H_2CO_3 => CaCO_3 + 2H_2O \tag{2}$$

It is not normally desirable for the carbonation to develop inward to the steel in reinforced concrete, because a reduced pH value breaks down the protective passivation layers of various iron oxides, which increases the risk of corrosion of the reinforcement. This is not a problem in unreinforced concrete or crushed concrete, as reinforcement corrosion does not take place.

The carbonation process is therefore most relevant and most desirable when reinforced concrete has been crushed after use in the building and the steel has been removed. There are many conditions that influence the amount of $CO_2$ the concrete binds from the air, such as: concrete quality, amount of cement, type of use, whether the product is indoors or outdoors, thickness, surface treatment, and whole concrete or crushed concrete. The larger the surface area that is in contact with the air, the more $CO_2$ will be carbonated in the concrete.

*6.2. Standards*

According to NS 3720:2018 [35], it is possible to include the effect of carbonation in GHG calculations for buildings, but it has not been common practice to do so to date. The effect of carbonation is calculated as given in NS-EN 16757 "Sustainable buildings— Environmental declarations—Product category rules for concrete and concrete elements" [58] and is included in modules B1, C3, C4, and D, depending on when in the life cycle the carbonation and absorption occurs. Various studies show the theoretical possibility of carbonation, both when the concrete product is in the building, and when the concrete product is crushed in the future, but as this depends on different and project-specific conditions, there are uncertainties in the figures.

NS 3720:2018 [35] does not specify the influence of DfD on carbonation, but, as crushing concrete increases carbonation, DfD would not be beneficial in this respect. On the contrary, DfD means that carbonation in module D (which is assumed to be greatest by crushing the concrete) is reduced. If concrete is reused in the future, corrosion of the reinforcement must be avoided, and the effect of carbonation must therefore be kept low. *Today's calculation rules do not provide figures on how much carbon is captured and stored by carbonation for different future uses of the concrete product.* Since the amount of carbon that is captured depends on the exposure of concrete to air and the time since the concrete was cast, it is theoretically possible to determine the extent of stored carbon both today and in different scenarios for use and DfD in the future.

## 7. Existing DfD Accounting Practices in the Norwegian Building Sector

The focus on building in a more circular way is strong in the Norwegian building sector and with Norwegian authorities. Pending any changes in future standards and regulations, there is a need to develop national frameworks that can account for and thus promote DfD, by making it quantifiable. Incentives for facilitating future dismantling and reuse will spur innovation for sustainable solutions, both financially and in terms of climate and resources.

An overview of the current "industry schemes" in the Norwegian building sector that have been developed to calculate and account for DfD in terms of life-cycle GHG emissions is given below.

The definitions of demountability and reusability are two important critical indicators that have not yet been assigned unambiguous criteria in current standards. In Norway, however, guidelines have been given for "reusability" in the program FutureBuilt ZERO [59] and in BREEAM-NOR v.6.0 [60]. These criteria should be further developed as a basis for how DfD should be credited as a potential GHG-reducing measure. Today, there are national schemes which, to some extent, promote DfD. These are briefly described below.

In BREEAM-NOR v.6.0 [60], Mat 07 Adaptability and reusability, criteria are given for "reusability", i.e., in practice, DfD. This is the minimum requirement to achieve an assessment of excellent and outstanding. The subject is also based on the EU's taxonomy for the environmental aspect of the CE. *In the BREEAM manual, guidelines are given for what must be satisfied for the project to be approved within this subject, but those measures are not linked to GHG analyses.*

New requirements are provided in the Norwegian building technical regulations TEK17 [13], § 9-5. Construction waste and reuse. According to this new provision, buildings must be designed and built so that they are arranged for later dismantling when this can be carried out within a practical and economically sound framework. *Here again, no guidelines have been given for how DfD should be considered in GHG calculations.*

FutureBuilt ZERO [59] encourages the reuse of building materials and can therefore potentially avoid combustion and substitution in the future production of new products. Up to 10% of the GHG emissions from module A1–A3 can be deducted for materials with documented reusability in module D. The condition for such a deduction is that "reusability" must be documented in accordance with FutureBuilt's criteria for circular buildings [59]. These criteria are approximately the same as those that apply to Mat 07 in BREEAM-NOR v.6.0 [60].

In the current version of the "Powerhouse Paris Proof", GHG emissions related to wood can be given a deduction of 100% when arranging for future dismantling and reuse (implying in practice "perpetual" reuse). The method is under development.

## 8. Conclusions and Future Work

*The mapping of standards on how GHG calculation are considered in DfD shows that there are no clear guidelines on that matter at present. In practice, this means that the GHG effect of facilitating the dismantling and reuse (DfD) of building components or materials in buildings when a building is to be disposed of (or rebuilt) is unclear. Additionally, today's standards do not provide*

*any incentives for facilitating DfD in the case of new construction. Future climate and resource gains may, therefore, be lost because of the limited knowledge of product solutions for DfD. This lack of knowledge can potentially both lead to increased design and construction costs.*

To address this knowledge issue and close the knowledge gap, the following issues—as listed below—should be addressed by an application in case studies, offering insightful inputs to the policy-making (e.g., standards) process:

- The allocation/distribution of future avoided emissions in the first, second, or third future building, including financial allocation (linked to the possible increased costs for the facilitation of DfD today).
- Including the importance of the number of reuses (number of buildings) for building products.
- The allocation of increased emissions from the facilitation of DfD today (more steel, increased durability, etc.) on the first, second, or future building.
- Could increasing the lifespan in module A1–A3 for reusable building products be an alternative to using module D for DfD?
- Which emission factors should be set for reusable products? There are usually no EPDs for reusable products at present.
- The time weighting (valuation) of future avoided emissions, seen in relation to the Paris Agreement's objective (reduction by 2050), as well as the uncertainty related to technology development (more climate-friendly materials, carbon capture in waste treatment, etc., for substituted building products) and likely reuse in the future.
- The treatment of biogenic carbon and carbonation in relation to DfD.
- The extent of GHG reductions by DfD vs. other GHG-reducing measures in GHG calculations (energy efficiency, use of more climate-friendly materials).
- Based on various analyses and the element of uncertainty, consider "innovation bonus" if there is too much uncertainty related to future GHG reductions in DfD.

**Author Contributions:** Conceptualization, C.L., M.T. and P.S.-M.; methodology, C.L., M.T., O.A.D., A.E. and P.S.-M.; writing—original draft preparation, C.L., O.A.D. and A.E.; writing—review and editing, C.L., M.T. and P.S.-M.; visualization, C.L., A.E. and P.S.-M.; supervision, C.L. and P.S.-M.; project administration, C.L., M.T. and P.S.-M.; funding acquisition, C.L., M.T. and P.S.-M. All authors have read and agreed to the published version of the manuscript.

**Funding:** This research was funded by the Research Council of Norway and several partners thought the "Sirkulære nybygg-Design og bygging for demontering og ombruk" project, with grant number 327777, and the Research Centre on Zero Emission Neighbourhoods in Smart Cities (FME ZEN), with grant number 257660.

**Institutional Review Board Statement:** Not applicable.

**Informed Consent Statement:** Not applicable.

**Data Availability Statement:** Not applicable.

**Conflicts of Interest:** The authors declare no conflict of interest.

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
