# Peer review of "Methods to Account for Design for Disassembly: Status of the Building Sector"

_buildings, doi:10.3390/buildings13041012_

Round 1

Reviewer 1 Report

The paper “ Methods to account for Design for Disassembly: Status for the 2 construction sector” provides an interesting review of DfD in the construction sector. This topic is highly interesting for the readers of the journal “Buildings”. However, before being considered for publication, the paper should be improved. The following comments would help to increase its quality:

·         Each statement should be backed up by a reference. For example, in line 46: “ the importance of "getting the  economics right" and the importance to act at the community level.

·         Include a reference for ISO 20887:2020

·         LCA can provided a broader environmental analysis than GHG, as more environmental impact categories can be taken into account. This should be clearly stated in the paper

·         Line 122: Which materials are more recycled? And which are usually sent to landfill?

·         Figure 1: Percentages are not easy to read

·         All abbreviations should be defined in the paper

·         Please, improve phrases such as “[34] have tested” “[35] have investigated

·         Subsection 4 does not fully fit with the rest of paper topic

·         Line 449: CO2 , 2 should be with subscript

·         The paper should show more DFD strategies and solutions, to put the whole paper into perspective. It is currently too focused on the methods to account, but would be greatly enhance including more information related to DfD

·         Many statements must be backed up by references, especially in a review paper. The same applies for all the different standards.

·         More focus regarding analysing only GHG versus a full LCA should be included in the paper

·         Maybe conclusions and future work could be shown in only one subsection

Author Response

Dear Reviewer 1,

Thank you very much for your comments and suggestions that helped us improve our manuscript. Please see our answer in the attached document. 

Kindly, Carine Lausselet, on behalf of the co-authors

Reviewer 2 Report

-           Some comments are provided below that could be helpful for the authors to improve their manuscript.

1.      The introduction is weak. The motivation for the research is missing.

2.     The author should highlight the research objectives of the DfD in the introduction separately. Also, how the authors would address their need to be mentioned.

3.     The manuscript lacks flow. The authors should make more efforts in presenting the work more systematically and clearly. The authors(s) have cited old citations throughout the manuscript.  The author(s) are also suggested to includes references from the latest publications (year 2022).

4.     The implications of Dfd are less developed. The authors should provide more insights on it, and its imapct on LCA.

5.     The findings of the study need to be more elaborate. This section needs to be developed and supported by previous work. The discussion needs to be improvised with a theoretical contribution.The findings of the discussion need to be strengthened with the previous research work.

6.     The discussion needs to be improvised with theoretical contribution.

7.     The conclusion is very weak. It should also be an extrapolation of the key findings from the research and not a summary. So, there should be conclusions around the background theory, data theory/analysis and, key outcomes. The authors should have included the following sub-sections within the conclusion section with more details:

Implications to theory and practice should be clearly stated;

 Key lessons learnt;

 Limitations of this research;

8.     The selection of the case location should be more elaborated.

9.  Proofread the whole manuscript as many typos and grammar errors are present.

10.   Future research directions should be improved; in that, they should stem from the awareness of the limitations and opening avenues related to the obtained outcomes

 11.  Author(s) should try to include some novel implications and unique contributions in the paper.

 12.  The Imapct of DfD on SDGs.

 13.  The mapping of regulations for how DfD takes into account GHG calculations shows that there are no clear guidelines. Could you suggest these guidelines??

Good Luck

Author Response

Dear Reviewer 2,

Thank you very much for your comments and suggestions that helped us improve our manuscript. Please see our answer in the attached document. 

Kindly, Carine Lausselet, on behalf of the co-authors

Reviewer 3 Report

The authors study the concept of DfD (design for disassembly) in the context of 3R policy (reduce, reuse, and recycle) at various phases (construction, use, and EoL)  of buildings. The legal guidelines on CHG's effect on disposal or rebuilding are argued.

Please provide the full form of the abbreviations at their first appearance.

The topic is very interesting and focused on one particular zone/geography; it is recommended to look into the policies adopted globally for recommendations.

Please elaborate the conclusions, Instead of future work, the authors can propose some policies for construction based on DfD principle to reduce the carbon emissions.

A bigger picture of a bill of materials (BOM) and their usage in percentages

Overall, it is a study work, the literature shall be expanded by some recent literature on DfD

Rios, Fernanda Cruz, Wai K. Chong, and David Grau. "Design for disassembly and deconstruction-challenges and opportunities." Procedia engineering 118 (2015): 1296-1304.

Anil Kumar, Gulivindala, et al. "A multi-layered disassembly sequence planning method to support decision making in de-manufacturing." Sādhanā 46.2 (2021): 102.

Crowther, Philip. "Re-valuing construction materials and components through design for disassembly." Unmaking Waste in Production and Consumption: Towards the Circular Economy. Emerald Publishing Limited, 2018.

Author Response

Dear Reviewer 3,

Thank you very much for your comments and suggestions that helped us improve our manuscript. Please see our answer in the attached document. 

Kindly, Carine Lausselet, on behalf of the co-authors

Round 2

Reviewer 1 Report

The authors have properly replied to all my comments. My final recommendation is Accept

Reviewer 2 Report

The authors have made all required comments and the paper is now ready for publication.

Good Work.